# Chemical Chaperone 4-PBA Mitigates Tumor Necrosis Factor Alpha-Induced Endoplasmic Reticulum Stress in Human Airway Smooth Muscle

**DOI:** 10.3390/ijms242115816

**Published:** 2023-10-31

**Authors:** Philippe Delmotte, Jane Q. Yap, Debanjali Dasgupta, Gary C. Sieck

**Affiliations:** Department of Physiology & Biomedical Engineering, Mayo Clinic, Rochester, MN 55905, USA; delmotte.philippe@mayo.edu (P.D.); yap.jane@mayo.edu (J.Q.Y.); dasgupta.debanjali@mayo.edu (D.D.)

**Keywords:** airway smooth muscle, chemical chaperone, endoplasmic reticulum stress, 4-phenylbutyric acid, unfolded protein

## Abstract

Airway inflammation and pro-inflammatory cytokines such as tumor necrosis factor alpha (TNFα) underlie the pathophysiology of respiratory diseases, including asthma. Previously, we showed that TNFα activates the inositol-requiring enzyme 1α (IRE1α)/X-box binding protein 1 spliced (XBP1s) endoplasmic reticulum (ER) stress pathway in human airway smooth muscle (hASM) cells. The ER stress pathway is activated by the accumulation of unfolded proteins in the ER. Accordingly, chemical chaperones such as 4-phenylbutyric acid (4-PBA) may reduce ER stress activation. In the present study, we hypothesized that chemical chaperone 4-PBA mitigates TNFα-induced ER stress in hASM cells. hASM cells were isolated from bronchiolar tissue obtained from five patients with no history of smoking or respiratory diseases. The hASM cells’ phenotype was confirmed via the expression of alpha-smooth muscle actin and elongated morphology. hASM cells from the same patient sample were then separated into three 12 h treatment groups: (1) TNFα (20 ng/mL), (2) TNFα + 4-PBA (1 μM, 30 min pretreatment), and (3) untreated control. The expressions of total IRE1α and phosphorylated IRE1α (pIRE1α^S724^) were determined through Western blotting. The splicing of *XBP1* mRNA was analyzed using RT-PCR. We found that TNFα induced an increase in pIRE1α^S724^ phosphorylation, which was mitigated by treatment with chemical chaperone 4-PBA. We also found that TNFα induced an increase in *XBP1s* mRNA, which was also mitigated by treatment with chemical chaperone 4-PBA. These results support our hypothesis and indicate that chemical chaperone 4-PBA treatment mitigates TNFα-induced ER stress in hASM cells.

## 1. Introduction

An underlying characteristic in the pathophysiology of airway diseases such as asthma, COVID-19, or chronic bronchitis is inflammation mediated by pro-inflammatory cytokines, such as tumor necrosis factor alpha (TNFα) [1,2,3,4,5]. We previously showed that the exposure of human airway smooth muscle (hASM) cells to TNFα results in an accumulation of damaged or misfolded proteins in the endoplasmic reticulum (ER), which then leads to the activation of an unfolded protein response (UPR^er^) (Figure 1) [6,7]. 

Binding immunoglobulin protein (BiP), also known as glucose-regulated protein 78 (GRP78), is a chaperone protein that acts as a sensor of protein unfolding. With the accumulation of unfolded proteins in the ER, BiP/GRP78 dissociates from sentinel proteins, leading to their autophosphorylation and downstream signaling (Figure 1) [6,7,8,9,10]. Three ER stress sentinel proteins involved are (1) inositol-requiring enzyme 1α (IRE1α), with autophosphorylation leading to the downstream splicing of X-box binding protein 1 (XBP1s), (2) protein kinase RNA-like endoplasmic reticulum kinase (PERK), with autophosphorylation leading to the phosphorylation of eukaryotic initiation factor 2 (eIF2), and (3) translocation to and cleavage of the activating transcription factor 6 (ATF6) within the Golgi [6,7,8,9,10]. These pathways trigger a homeostatic signaling cascade to restore normal function, such as halting protein translation or increasing the production of chaperone proteins, for example. In previous studies, we found that TNFα activates only the pIRE1α/XBP1s ER stress pathway in hASM cells [6,7].

Chemical chaperones such as 4-phenyl butyric acid (4-PBA) promote protein refolding and thus potentially inhibit ER stress. The therapeutic use of chemical chaperone 4-PBA has been explored in a variety of ER stress-associated diseases [11,12,13,14,15]. The aim of this study is to explore the potential of chemical chaperone 4-PBA to mitigate ER stress in hASM cells. In the present study, we hypothesize that treatment with chemical chaperone 4-PBA will mitigate the TNFα-induced pIRE1α/XBP1s ER stress pathway in hASM cells (Figure 1). 

## 2. Results

### 2.1. TNFα Induces Phosphorylation of IRE1α in hASM Cells

Using Western blotting, we found that the ratio of pIRE1α^S724^ to total IRE1α was increased after 12 h TNFα treatment compared to the untreated time-matched control groups (*n* = 5, ** p* < 0.05) (Figure 2A–C and Appendix A). The total IRE1α protein expression level was comparable between the 12 h TNFα-treated group and the untreated time-matched control group for each patient (Figure 2B). The mean percent change from the control for the ratio of pIRE1α^S724^ to total IRE1α in the 12 h TNFα-treated groups was ~50%, with values ranging from 17 to 115% (Figure 2D). 

### 2.2. Chemical Chaperone 4-PBA Mitigates TNFα-Induced Phosphorylation of IRE1α in hASM Cells

Using Western blotting, we found that the ratio of pIRE1α^S724^ to total IRE1α was decreased after 12 h 4-PBA + TNFα treatment compared to the 12 h TNFα groups (*n* = 5, ** p* < 0.05) (Figure 2A–C). The total IRE1α protein expression level was comparable between the 12 h 4-PBA + TNFα-treated group and the 12 h TNFα treatment for each patient (Figure 2B). The mean percent change from the TNFα-treated groups to the TNFα + 4-PBA-treated groups for the ratio of pIRE1α^S724^ to total IRE1α was ~50% (*n* = 5, * *p* < 0.05) (Figure 2D). A significant decrease in the ratio of pIRE1α^S724^ to total IRE1α was also observed in the 12 h 4-PBA + TNFα treatment groups compared to the untreated time-matched control groups (*n* = 5, * *p* < 0.05) (Figure 2C and Appendix A).

### 2.3. TNFα Induces Splicing of XBP1 mRNA in hASM Cells

Using PCR, we found that the ratio of XBP1s to XBP1u was increased after 12 h TNFα treatment when compared to the untreated time-matched control groups (Figure 3A–C) (*n* = 5, * *p* < 0.05). The XBP1u mRNA level was comparable between the 12 h TNFα-treated group and the untreated time-matched control group for each patient (Figure 3B). The mean percent change from the control for the ratio of XBP1s to XBP1u in the TNFα-treated groups was ~50%, with values increasing by ~200% (*n* = 5, * *p* < 0.05) (Figure 3D). 

### 2.4. Chemical Chaperone 4-PBA Mitigates TNFα-Induced Splicing of XBP1 mRNA in hASM Cells

Using PCR, we found that the ratio of XBP1s to XBP1u was decreased after the 12 h 4-PBA + TNFα treatment compared to 12h TNFα treatment (*n* = 5, * *p* < 0.05) (Figure 3A–C). The ratio of XBP1s to XBP1u after 12 h 4-PBA + TNFα treatment was comparable to the untreated time-matched control (Figure 3A–C). The XBP1u mRNA level was decreased after 12 h 4-PBA + TNFα treatment compared to 12 h TNFα treatment (Figure 3B). The mean percent change from the TNFα-treated groups to the TNFα + 4-PBA-treated groups was ~50% for the ratio of XBP1s to XBP1u (*n* = 5, * *p* < 0.05) (Figure 3D). 

### 2.5. 4-PBA Alone Has No Effect on Phosphorylation of IRE1α and XBP1 Splicing in hASM Cells

In selected experiments, we examined the effect of 4-PBA treatment alone on the ratio of pIRE1α^S724^ to total IRE1α and the ratio of XBP1s to XBP1u. We found that treatment with chemical chaperone 4-PBA alone had no effect (Appendix A).

## 3. Discussion

The results of the present study show that exogenous chemical chaperone 4-PBA is effective in mitigating the unfolded protein response and activation of the pIRE1α^S724^/*XBP1s* ER stress pathway induced by TNFα treatment in hASM cells. This was reflected by a decrease in TNFα-induced pIRE1α^S724^ phosphorylation and the subsequent splicing of *XBP1s*. Treatment with chemical chaperone 4-PBA alone showed no effect on pIRE1α^S724^ or *XBP1s.*

### 3.1. TNFα Induces Phosphorylation of IRE1α and mRNA Splicing of XBP1 at 12 h

Inflammation is a key component of airway diseases and has been shown to induce ER stress in many cell types, including hASM cells [3,6,7,16,17,18,19,20]. Consistent with our previous studies, we showed that pro-inflammatory cytokine TNFα induces ER stress in hASM cells, reflected by the activation of the pIRE1α^S724^/XBP1s pathway [6,7]. Interestingly, pro-inflammatory cytokine TNFα does not activate the PERK/eIF2α or the ATF6 ER stress pathway [6,7]. Previously, we showed that TNFα treatment induces pIRE1α^S724^ and subsequent *XBP1s* at 6 h with maximal pIRE1α^S724^ at 12 h [7]. IRE1α contains two enzymatic activities, a kinase and an endoribonuclease (RNase), both located on the cytosolic side [21,22,23]. Upon sensing ER stress, IRE1α oligomerizes, which juxtaposes the kinase domains, leading to trans-autophosphorylation, and activates the RNase activity of pIRE1α^S724^, leading to the splicing of XBP1 mRNA [21,22,23]. XBP1s has been shown to transcriptionally activate a multitude of target genes [21,22,23,24,25,26,27] and could potentially affect hASM cells’ contractility and/or proliferation. The protein kinase activity of pIRE1α^s724^ has been reported in several studies, but its role is still under scrutiny [21,28,29]. In the current study, we confirmed that, in hASM cells, TNFα treatment increases the phosphorylation of IRE1α as well as increasing the mRNA splicing of *XBP1* at 12 h. We also found that TNFα treatment has no effect on total IRE1α protein expression at 12 h. This result indicates that TNFα treatment does not increase pIRE1α^S724^ through an increase in total IRE1α protein expression. 

### 3.2. 4-PBA Mitigates ER Stress in hASM Cells

The chemical chaperone 4-PBA is a low-molecular-weight fatty acid commonly considered an ER stress inhibitor [30,31,32]. The hydrophobic regions of chemical chaperone 4-PBA interact with the exposed hydrophobic parts of unfolded proteins, promoting the refolding of these proteins and reducing protein accumulation, thus reducing ER stress [30,31,32]. Previous studies have shown that chemical chaperone 4-PBA is effective in reducing ER stress induced by inflammation in various cell types and diseases, such as urea cycle disorders, cystic fibrosis, malignant gliomas, or motor neuron diseases. [31,33,34]. However, the potential effect of chemical chaperone 4-PBA in hASM cells has not been explored. In the present study, we examined the effect of chemical chaperone 4-PBA on ER stress induced by pro-inflammatory cytokine TNFα in hASM cells. We found that pretreatment with chemical chaperone 4-PBA mitigated the TNFα-induced phosphorylation of IRE1α as well as the increase in the mRNA splicing of *XBP1*, thus reducing the TNFα-induced ER stress in hASM cells. Importantly, the treatment of hASM cells with only chemical chaperone 4-PBA has no effect on pIRE1α^S724^ or *XBP1s*.

### 3.3. Study Limitations

The number of patients was determined through a power analysis of the primary outcome measures. The patient sample size was small, but it is unlikely that these five patients were outliers. However, further experiments will be needed to both confirm the effect of chemical chaperone 4-PBA on TNFα-induced ER stress and confirm that chemical chaperone 4-PBA has no effect on its own, before any clinical study can be contemplated. While patients did not have a history of chronic lung disease, asthma, any other respiratory disease, or smoking, they were undergoing lung surgery, most likely related to cancer resection. Tissue samples were evaluated by a clinical pathologist and only “normal” lung tissue was used in this study. We observed a large variance in the phosphorylation of IRE1α as well as in the mRNA splicing of *XBP1*, which may be a reflection of the age of, or medications taken by, the patients. The effect of age on TNFα-induced ER stress and the efficacy of chemical chaperone 4-PBA is of great interest, but is beyond the scope of this study. 

### 3.4. Clinical Significance

Chemical chaperone 4-PBA is a United States Food and Drug Administration-approved drug which has been tested as a potential therapeutic agent in patients with urea cycle disorders, cystic fibrosis, malignant gliomas, or motor neuron diseases, amongst other conditions. The role of ER stress in airway diseases such as asthma or COPD is still under investigation. In the present study, we show that pro-inflammatory cytokine TNFα induced ER stress and chemical chaperone 4-PBA mitigated TNFα-induced ER stress in hASM cells, showing a potential future therapeutic application of chemical chaperones in airway diseases.

## 4. Material and Methods

### 4.1. Experimental Design

#### 4.1.1. Patient Samples

Mayo Clinic’s Institutional Review Board (IRB #16-009655) reviewed the research protocol and determined no further review was required due to minimal risk to patients for the following reasons: (1) patient anonymity was maintained and no patient identifiers were stored when collecting the tissue, although patient history was recorded, including sex, demographics, pulmonary disease status, pulmonary function testing, imaging, co-morbidities, and medications; (2) patient samples were numbered without the possibility of identifying patients; (3) patient consent was obtained during pre-surgical evaluation in a non-threatening environment. Samples from 5 patients (3 males, 2 females) were selected. Patients ranged in age from 34 to 75 years, without a history of chronic lung disease, asthma, or any other respiratory disease, or a history of smoking (Figure 4, Table 1). However, patients were undergoing lung surgery, most likely related to cancer resection. Importantly, all tissue was evaluated through clinical pathology and only “normal” lung tissue was provided for this study. 

#### 4.1.2. Dissociation of Cells from Bronchiolar Tissue Samples

During lung surgery, samples of third- to sixth-generation normal (non-diseased) bronchiolar tissue were obtained. After pathological evaluation and determination of normal tissue, the smooth muscle layer was dissected. Cells were then dissociated using papain and collagenase with ovomucoid/albumin separation as per manufacturer instructions and as previously described (Worthington Biochemical, Lakewood, NJ, USA) [7,35,36,37,38]. Cells were maintained in phenol red-free DMEM/F-12 medium (Invitrogen, Carlsbad, CA, USA), supplemented with 10% fetal bovine serum (Cat. No. A3840002, Gibco, Thermo Fisher Scientific, Rockford, IL, USA) and 100 U/mL penicillin/streptomycin culture dishes at 37 °C, 5% CO_2_—95% air. Only passages 1–3 were used in our study (Figure 4). 

#### 4.1.3. Confirmation of hASM Phenotype

The phenotype of the hASM cells was confirmed through immunocytochemical analysis of the expression of α-smooth muscle actin (α-SMA), as previously described [39] (Figure 5). Additionally, cells expressing α-SMA were found to be larger, with an elongated shape (Figure 5). Cells were plated at a density of ~10,000 cells per well in a Nunc™ Lab-Tek™ II Chamber 8-well multi-chamber slide (Thermo Fisher Scientific, Rockford, IL, USA), then fixed with 4% paraformaldehyde in 1X phosphate-buffered saline (PBS) for 10 min (room temperature) and finally washed with 1X PBS. Cells were blocked using an antibody diluent solution containing 10% normal donkey serum (Sigma-Aldrich, St. Louis, MO, USA), 0.2% triton X-100, and 1X PBS, incubated overnight at 4 °C with anti-α-SMA antibody (ab5694, Abcam, Boston, MA, USA) at a dilution of 1:500, and then incubated for 1 h with donkey anti-rabbit biotin-conjugated secondary antibody at 1:400 concentration (Jackson Immunoresearch, West Grove, PA, USA), followed by Streptavidin Alexa Fluor 568 (1:200 in PBS; Invitrogen, Carlsbad, CA, USA). Cells were then mounted using Fluoro-Gel II medium with 4′,6-Diamidino-2-Phenylindole and Dihydrochloride (DAPI) (Cat. No. 17985-50 Electron Microscopy Sciences, Hatfield, PA, USA), and imaged using a Nikon Eclipse A1 laser scanning confocal microscope with a ×60/1.4 NA oil-immersion objective at 12-bit resolution into a 1024 × 1024-pixel array (Nikon Instruments Inc., Melville, NY, USA). Cells expressing α-SMA (hASM cells) with an elongated shape accounted for ~95% of all dissociated cells (Figure 4 and Figure 5).

#### 4.1.4. Treatment Groups

Prior to experimentation, cells were serum-deprived for 48 h, and then cells from each patient were assigned to one of three groups: (1) 12 h TNFα-treated (TNFα, 20 ng/mL; Cat. No. T6674, Sigma Aldrich, St. Louis, MO, USA), (2) 12 h TNFα + 4-PBA-treated (1 μM, 30 min pretreatment, CAS. No: 1716-12-7, Tocris Bioscience, Bristol, UK), and (3) 12 h time-matched untreated control (Figure 4). The concentration (20 ng/mL) and exposure time (12 h) for the TNFα treatment were based on our previous study showing that the maximal phosphorylation of pIRE1α^S724^ and splicing of *XBP1* were achieved under those conditions [7]. In selected experiments, we examined the effect of 4-PBA treatment alone on pIRE1α^S724^ and the splicing of *XBP1* (Appendix A). All experiments were in duplicate.

### 4.2. Determining pIRE1α^S724^ and XBP1s ER Stress Response 

#### 4.2.1. pIRE1α^S724^ and Total IRE1α Protein Expression

hASM cells were lysed using 1X Cell Lysis Buffer (Cat. No. 9803, Cell Signaling Technology, Danvers, MA, USA) containing protease (Cat. No. 11836170001, Roche, Burlington, MA, USA) and phosphatase inhibitors (PhosSTOP, Cat. No. 4906845001, Roche). Protein samples were extracted and the protein concentration was determined using a Lowry assay (Bio-Rad, Berkeley, CA, USA), following manufacturer protocol. For each Western blot, 100 µg total protein was denatured in 1X Laemmli sample buffer (Bio-Rad) with 5% β-mercaptoethanol at 95 °C for 5 min, separated by stain-free SDS-PAGE (Bio-Rad), and transferred to a polyvinylidene difluoride (PVDF) membrane (Bio-Rad) using a Trans-Blot Turbo system (Bio-Rad). The protein levels of total IRE1α (NB100-2324, Novus Biologicals, Littleton, CO, USA) and pIRE1α^S724^ (ab124945, Abcam) were detected using primary antibodies at a dilution of 1:1000. The antibodies for total IRE1α and pIRE1α^S724^ were validated in a previous study [7]. The total protein in each lane was visualized using the ChemiDoc Imaging system (Bio-Rad) and analyzed using Image Lab software version 6.0.1. Band intensity corresponding to total IRE1α or pIRE1α^S724^ was normalized to the total protein in the gel.

#### 4.2.2. Measuring the Splicing of XBP1 mRNA

The total RNA from the hASM cells was extracted using an RNeasy extraction kit (Cat. No. 74104, Qiagen, Hilden, Germany) according to manufacturer instructions and quantified using a Nano Spectrophotometer (Thermo Fisher Scientific). A total of 500 ng mRNA was used for complementary DNA (cDNA) synthesis. Subsequently, traditional PCR and quantitative real-time qPCR were conducted using a LightCycler 480 SYBR Green I Master (Cat. No. 04707516001, Roche) to estimate the mRNA expression of XBP1s and XBP1u. The primers for XBP1s are 5′-TCTGCTGAGTCCGCAGCAGG-3′ for XBP1s-F and 5′-CTCTAAGACTAGAGGCTTGG-3′ for XBP1s-R. The primers for XBP1u are 5′-CAGACTACGTGCGCCTCTGC-3′ for XBP1u-F and 5′-CTTCTGGGTAGACCTCTGGG-3′ for XBP1u-R. The samples were then separated in a Tris Borate EDTA (TBE) agarose gel to validate and quantify the expression of XBP1s and XBP1u using a ChemiDoc Imaging system (Bio-Rad). 

### 4.3. Statistical Analysis

Cells from bronchiolar samples from five patients were used for each experimental protocol (Table 1). Cells from female and male patients were evaluated and sex was considered a random variable. From the same patient, cells were divided into three groups for within-subject comparisons. The Shapiro–Wilk test was performed to confirm normal distribution and a power analysis was conducted using the preliminary results to determine the number of patient samples for both Western blot and PCR analyses. A two-way ANOVA was performed for statistical analyses using GraphPad Prism 9. If justified through ANOVA, a Bonferroni post hoc test was used to compare across groups. All data represent mean ± SEM, and each color represents data from one patient. Significant differences are indicated by * (*p* < 0.05).

## 5. Conclusions

Chemical chaperones such as 4-PBA have been extensively used to mitigate ER stress in several disease models. In airway diseases, the role of ER stress induced by inflammation is still under investigation. The present study shows that chemical chaperone 4-PBA mitigates TNFα-induced ER stress in hASM cells, showing a potential future therapeutic application in airway diseases.

## Figures and Tables

**Figure 1 ijms-24-15816-f001:**
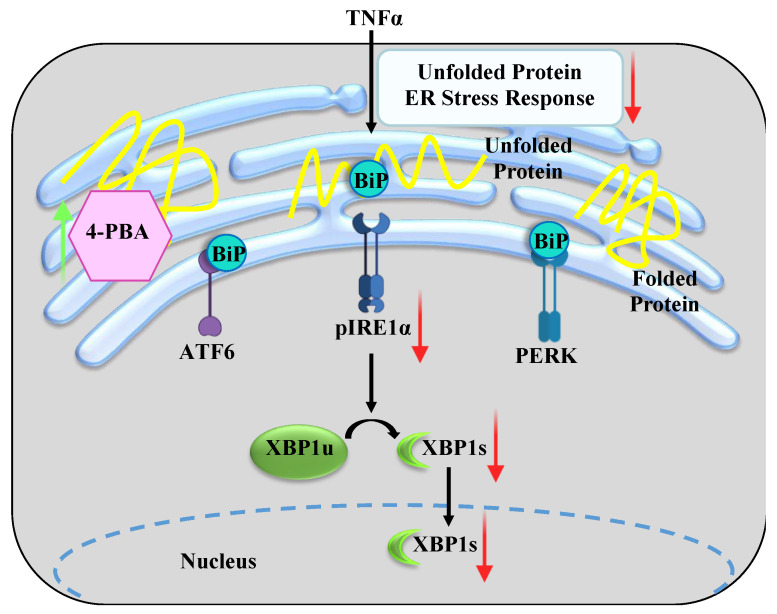
Conceptual framework of this study. Tumor necrosis factor alpha (TNFα) induces the accumulation of unfolded proteins (unfolded and folded proteins represented in yellow) in the endoplasmic reticulum (ER), causing the dissociation of binding immunoglobulin protein (BiP) from sentinel proteins and triggering an ER stress response. In human airway smooth muscle (hASM) cells, TNFα induces phosphorylation of inositol-requiring enzyme 1α (IRE1α) at serine residue 724 with downstream alternative splicing of X-box binding protein 1 (XBP1) mRNA. In the present study, we tested the hypothesis that promoting the refolding of proteins using chemical chaperone 4-phenylbutyric acid (4-PBA) will mitigate the ER stress response.

**Figure 2 ijms-24-15816-f002:**
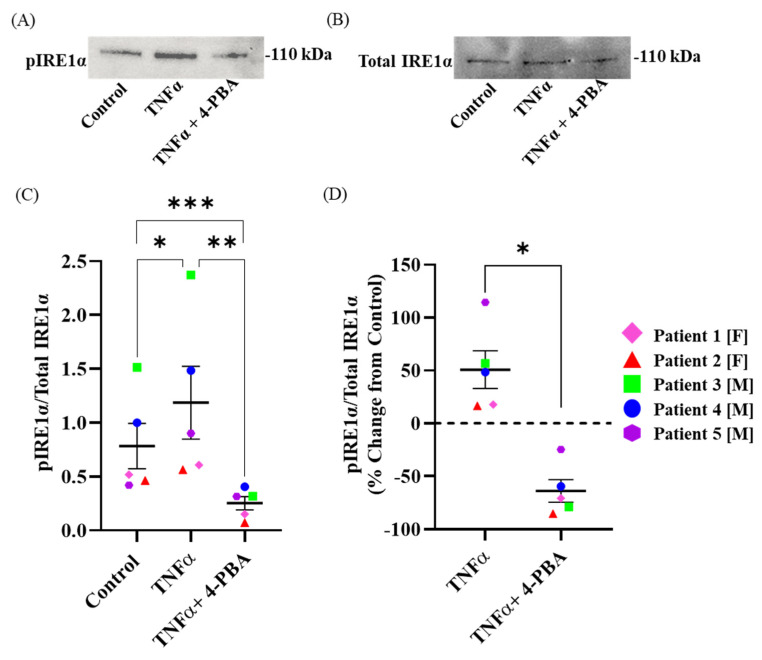
Chemical chaperone 4-PBA mitigates the increase in pIRE1α^S724^ induced by TNFα exposure in hASM cells. Representative Western blots of (**A**) pIRE1α^S724^ (ab124945) and (**B**) total IRE1α (NB100-2324) showing in increase in pIRE1α^S724^ after TNFα exposure, which is mitigated by treatment with chemical chaperone 4-PBA. In each patient, the ratio of pIRE1α^S724^ to total IRE1α (**C**) is increased after TNFα exposure compared to untreated control hASM cells and is summarized in a box–whisker plot with each patient represented by different colors. The ratio of pIRE1α^S724^ to total IRE1α, expressed as % change from untreated control hASM cells (**D**), is summarized in a box–whisker plot and shows an increase in the pIRE1α^S724^/total IRE1α ratio after TNFα exposure in each patient, which is mitigated by chemical chaperone 4-PBA. Statistical analyses were performed using a two-way ANOVA (*, ** and *** *p* < 0.05; *n* = 5 per treatment group).

**Figure 3 ijms-24-15816-f003:**
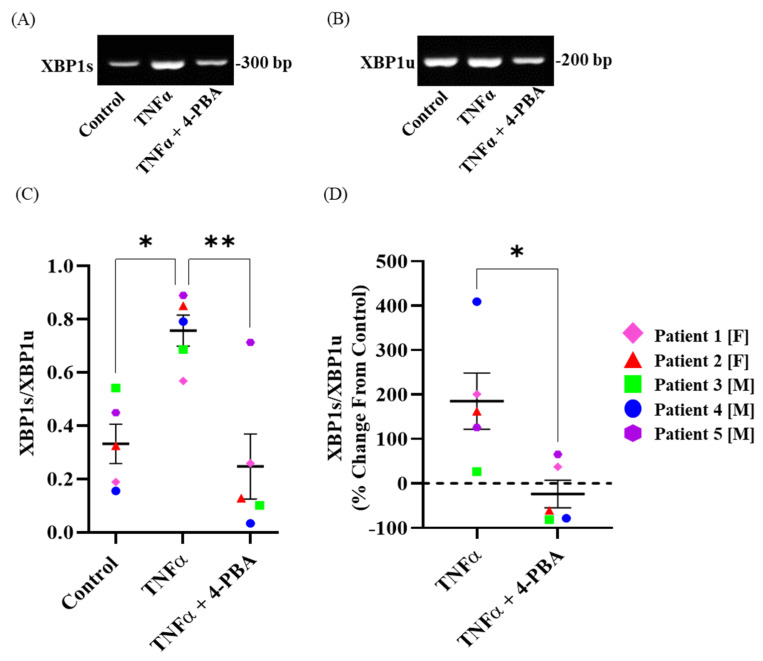
Chemical chaperone 4-PBA mitigates the increase in *XBP1s* induced by TNFα exposure in hASM cells. Representative PCR agarose gel images for *XBP1s* (**A**) and *XBP1u* (**B**), showing an increase in *XBP1*s after TNFα exposure that is mitigated by chemical chaperone 4-PBA treatment. In each patient, the ratio of *XBP1s* to *XBP1u* mRNA (**C**) is increased after TNFα treatment compared to untreated control hASM cells, summarized in a box–whisker plot with each patient represented by different colors. The ratio of *XBP1s* to *XBP1u* mRNA expressed as % change from untreated control hASM cells (**D**) is summarized in a box–whisker plot, showing an increase in the *XBP1s*/*XBP1u* ratio after TNFα exposure in each patient that is mitigated by chemical chaperone 4-PBA. Statistical analyses were performed using a two-way ANOVA (* and ** *p* < 0.05; for each measure, *n* = 5 per treatment group).

**Figure 4 ijms-24-15816-f004:**
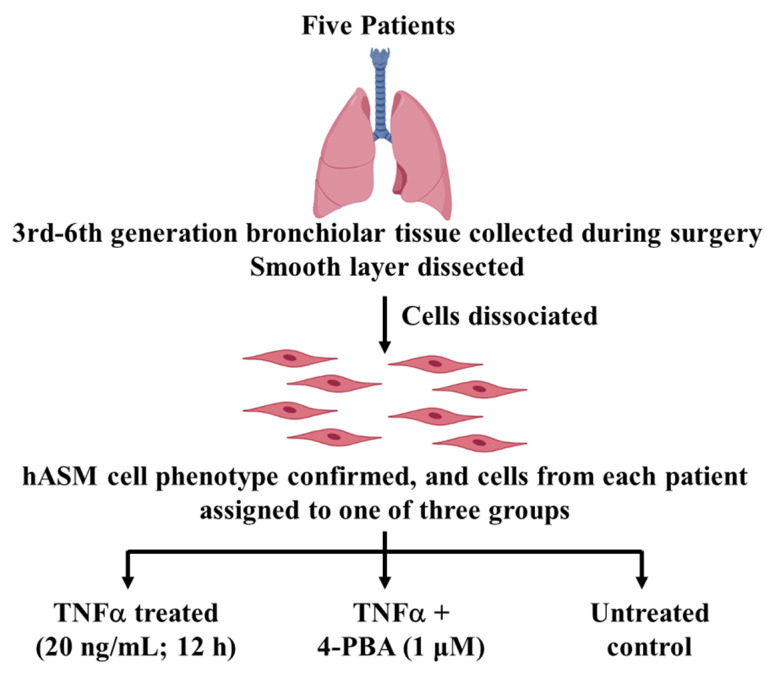
Experimental design. Third- to sixth-generation bronchiolar samples were obtained from 5 patients ranging in age from 30–75 years. The smooth muscle layer was dissected, and cells dissociated. Cells were phenotyped based on expression of alpha-smooth muscle actin (α-SMA) and morphologically distinct elongated shape. hASM cells from the same patient were divided into three groups and treated for 12 h: (1) TNFα-treated (20 ng/mL); (2) TNFα + 4-PBA-treated (1 μM; 30 min pretreatment prior to TNFα); and (3) untreated time-matched control.

**Figure 5 ijms-24-15816-f005:**
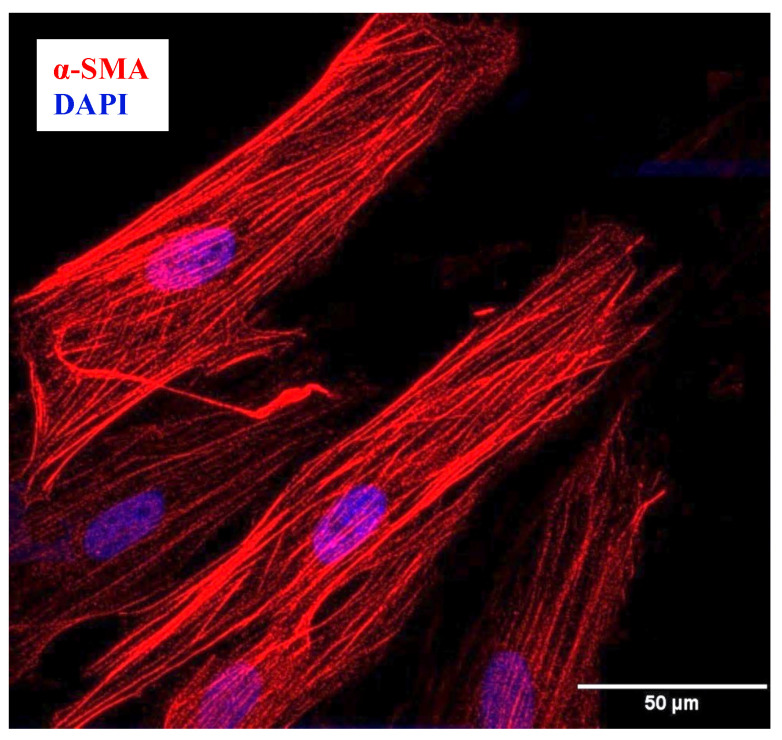
Phenotyping of dissociated hASM Cells. Representative Z projection image of hASM cells displaying immunoreactivity for α-SMA expression (red channel). In addition, hASM cells showed a morphologically distinct elongated shape (scale bar = 50 μm).

**Table 1 ijms-24-15816-t001:** Patient demographics.

Patient No.	1	2	3	4	5
Sex	F	F	M	M	M
Age (years)	64	61	71	75	34
Asthma	No	No	No	No	No
COPD	No	No	No	No	No
Pulmonary Fibrosis	No		No	No	No
Pulmonary Hypertension	No	No	No	No	No

## Data Availability

All data are presented in the main manuscript and Appendix A.

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
