# Peer review of "Chemical Chaperone 4-PBA Mitigates Tumor Necrosis Factor Alpha-Induced Endoplasmic Reticulum Stress in Human Airway Smooth Muscle"

_ijms, 2023, doi:10.3390/ijms242115816_

Round 1
Reviewer 1 Report
Comments and Suggestions for Authors
The main question addressed by this article is whether 4-phenylbutyric acid (4-PBA) can mitigate endoplasmic reticulum (ER) stress induced by tumor necrosis factor alpha (TNFα) in human airway smooth muscle (hASM) cells. It focuses on the impact of 4-PBA on ER stress and its potential as a treatment for respiratory diseases, particularly asthma, which is associated with airway inflammation and pro-inflammatory cytokines
As for originality, while the study builds upon existing knowledge of ER stress and its role in respiratory diseases, it doesn't introduce a radically novel concept. However, it addresses a specific research question within the field by exploring the potential of 4-PBA to mitigate ER stress in hASM cells. This can be seen as a valuable contribution to the ongoing efforts to better understand and manage respiratory diseases, potentially by targeting ER stress pathways. In this sense, it does address a specific gap in the field by examining the effects of 4-PBA in this context.
The conclusions are consistent with the evidence and arguments presented and they address the main question posed.
The references are appropriate.
I read this manuscript with great interest.
Authors carried out a well-written and well-designed study, although a few points raised some concerns (please see below):
- Please clearly specify the aim of the study
- Please clearly state the inclusion and exclusion criteria
- IF lung tissue was obtained by patients with no respiratory diseases, why these subjects underwent lung surgery? This point is not clear and should be better addressed.
- In discussion section, please had a subchapter with study limitations, first of all the very small sample size.
Author Response
General comment: The main question addressed by this article is whether 4-phenylbutyric acid (4-PBA) can mitigate endoplasmic reticulum (ER) stress induced by tumor necrosis factor alpha (TNFα) in human airway smooth muscle (hASM) cells. It focuses on the impact of 4-PBA on ER stress and its potential as a treatment for respiratory diseases, particularly asthma, which is associated with airway inflammation and pro-inflammatory cytokines. As for originality, while the study builds upon existing knowledge of ER stress and its role in respiratory diseases, it doesn't introduce a radically novel concept. However, it addresses a specific research question within the field by exploring the potential of 4-PBA to mitigate ER stress in hASM cells. This can be seen as a valuable contribution to the ongoing efforts to better understand and manage respiratory diseases, potentially by targeting ER stress pathways. In this sense, it does address a specific gap in the field by examining the effects of 4-PBA in this context.
The conclusions are consistent with the evidence and arguments presented and they address the main question posed.
The references are appropriate.
I read this manuscript with great interest.
Response: We thank the reviewer for the summary of our results and the positive comments.
Comment 1: Authors carried out a well-written and well-designed study, although a few points raised some concerns (please see below):
- Please clearly specify the aim of the study
Response: We revised the manuscript to clearly specify the aim of the study.
Comment 2: Please clearly state the inclusion and exclusion criteria
Only used bronchial tissue from patient with no history of chronic lung disease, asthma, any other respiratory disease, or history of smoking. After cell dissociation, we confirmed that more than 90% of the cells were ASM cells, otherwise they were excluded from this study.
Comment 3: IF lung tissue was obtained by patients with no respiratory diseases, why these subjects underwent lung surgery? This point is not clear and should be better addressed.
Response: Patients were undergoing lung surgery, most likely related to cancer resection. Importantly, all tissue was evaluated by clinical pathology and only “normal” lung tissue was provided for this study. We addressed this point in our revised manuscript.
Comment 4: In discussion section, please had a subchapter with study limitations, first of all the very small sample size.
Response: we agree with the reviewer that the sample size is small and we added a subchapter to discuss the study limitations.
Reviewer 2 Report
Comments and Suggestions for Authors
In this paper Delmotte and collaborators tested the hypothesis that promoting the refolding of proteins using chemical chaperone 4-phenylbutyric acid (4-PBA) will mitigate the ER stress response.
1. table 1: the rows corresponding to age and sex are reversed. Furthermore, from the table we can see that the age range is between 34 and 75 years (in the text 30-75).
2. Sex was not reported in table 1. However, this information is present in the manuscript. Please add the information in the table.
3. Lines 206-207: information already present in the text, there is no need to add it as a table caption
4. methods do not report whether experiments were performed in duplicate/triplicate. This information would be helpful.
Author Response
General comment: In this paper Delmotte and collaborators tested the hypothesis that promoting the refolding of proteins using chemical chaperone 4-phenylbutyric acid (4-PBA) will mitigate the ER stress response.
Response: We thank the reviewer for the succinct summary of our study.
Comment 1: table 1: the rows corresponding to age and sex are reversed. Furthermore, from the table we can see that the age range is between 34 and 75 years (in the text 30-75).
Response: We revised the manuscript and the table accordingly.
Comment 2: Sex was not reported in table 1. However, this information is present in the manuscript. Please add the information in the table.
Response: we added this information in the revised table.
Comment 3: Lines 206-207: information already present in the text, there is no need to add it as a table caption.
Response: we removed the table caption accordingly.
Comment 4: methods do not report whether experiments were performed in duplicate/triplicate. This information would be helpful.
Response: we revised the manuscript to include that we performed duplicate.
Round 2
Reviewer 1 Report
Comments and Suggestions for Authors
Ok to accept now